# Pitch characteristics of real-world infant-directed speech vary with pragmatic context, perceived adult gender, and infant gender

**Emily M. Neer** [1]*, **Anvi Brahmbhatt** [2], **Catherine R. Walsh** [1], **Anne S. Warlaumont** [2]

**1** Department of Psychology, University of California, Los Angeles, United States of America,
**2** Department of Communication, University of California, Los Angeles, United States of America

* emily.neer.7@gmail.com

## Abstract

Children's everyday language environments can be full of rich and diverse input, especially adult speech. Prosodic modifications when adults speak to infants are observed cross-culturally and are believed to enhance infant learning and emotion. However, factors such as what and why adults are speaking as well as speaker gender can affect the prosody of adults' speech. This study asks whether prosodic modifications to infant-directed speech depend on perceived adult speaker gender, assigned infant gender, and the perceived pragmatic function of an utterance. We examined 3,607 adult speech clips from daylong home audio recordings of 60 North American, English-speaking, 3- to 20-month-old infants (28 female). Adult speakers used significantly more imperatives and questions and sang more frequently to infants than other adults. While infant-directed speech tended to have greater mean pitch and pitch modulation than adult-directed speech overall, these patterns were modulated, sometimes in complex ways, by pragmatic function, perceived adult gender, and infant gender. For example, we found that female-sounding adult speakers exhibited greater IDS-ADS mean pitch differences than male-sounding adult speakers when providing information or engaging in conversational niceties. An additional example is that male-sounding adults used higher pitch when singing to male infants compared to female infants. These findings invite further research on how individual, demographic, and situational factors affect speech to infants and possibly infant learning. The study's pragmatic context tags are added to an existing open dataset of infant- and adult-directed speech.

## Introduction

Adults speak differently to infants than to other adults. This phenomenon is established across cultures [1–3]. Infant-directed speech (IDS) often differs from adult-directed speech (ADS) in having higher pitch, greater variation in intonation and other

**Data availability statement:** The data and code to reproduce the analyses presented here are publicly accessible on GitHub: https://github.com/emucla/context-study. Analyses were also pre-registered. Materials and the preregistration for this research are available at the following OSF page: https://osf.io/va7c8/

**Funding:** A.S.W: National Science Foundation grants (1529127 and 1539129/1827744) https://www.nsf.gov/ A.S.W: James S McDonnell Foundation Scholar Award https://grants.jsmf.org/220020507/ Neither funders played a role in study design, data collection and analysis, decision to publish, or preparation of the manuscript.

**Competing interests:** The authors have declared that no competing interests exist.

prosodic features [3–5], elongated vowels [6], shorter utterances [7], and simpler lexicons [8]. These differences are believed to have benefits for children's attention to and acquisition of language [9–11].

The focus of the current study is on voice pitch across IDS and ADS registers. Although some speakers and some cultures manipulate IDS acoustics more than others, generally speaking, adults tend to use overall higher pitch [2] and more pitch modulations when speaking to infants compared to adults [3]. It has been theorized that higher pitch may help infants to match adult speech acoustics to their own, facilitating comparison between self-produced and caregiver-produced utterances [12]. Others have emphasized that greater pitch variability within an utterance may make it more attention-grabbing or help to highlight key words to facilitate infant language learning [13].

It is important to note that much of the research on acoustic differences between IDS and ADS is limited to speech samples produced in specific contexts, with most of the focus being on the speech that is intended to engage an infant's interest and facilitate language–learning. Moreover, speech samples are often obtained through elicitation by a researcher. In contrast, infants often experience infant-directed speech that has a range of different goals, for example, that of soothing, rather than exciting, and of perhaps facilitating sleep or cessation of an activity, rather than engagement in conversation or with objects. In addition, the ways in which adults speak in real-world settings, such as in their homes while interacting with multiple family members and engaged in various day-to-day tasks and facing various stressors, have the potential to greatly alter the acoustic features of both ADS and IDS.

### Pragmatic contexts and acoustic characteristics

Despite general trends differentiating IDS from ADS registers, there is also research indicating that pragmatics, or the functional use of speech, affects speech acoustics in ways that may modulate the IDS-ADS differences. For infant-directed speech, pragmatic contexts can include attention-bidding, approval, prohibition, soothing, and singing [14], among others. Prior work has demonstrated that pitch can vary across pragmatic contexts. For example, mothers are more likely to use lower pitch when soothing an infant [14] and higher pitch when trying to get an infant's attention [14,15]. Research has also found that adults use greater pitch variability when arousing or entertaining infants, but less variable pitch when soothing infants [16,17]. When singing, parents tend to use a higher pitch [3,18,19]. Moreover, different pragmatic types of singing have different acoustic features: Trainor et al. found that infant play songs were rated to have greater pitch variability than infant lullabies [18]. Pitch differences are also associated with pragmatics (e.g., agreement, criticism, suggestion) in adult speech to other adults [20].

### Effects of adult and infant gender on communication

Another factor that can strongly influence adult voice acoustics is gender. Adult females generally have higher voice pitch compared to adult males [21,22] as well as other acoustic differences. These speaker gender differences in voice acoustics

are in part attributable to anatomical differences between males and females in vocal folds and the vocal tracts associated with speech production [21,23]. In addition to anatomical differences, sociocultural factors contribute to differences in voice acoustics between males and females [21]. For example, a speaker's emotional state, race, sex, and age can also affect acoustic variation [24] as well as social and cultural influences [25]. Variation in speech thus carries cues to attributes and states, such as gender, of the speaker, making it socially meaningful [26].

Adult gender is also associated at the group level with differences in the interactions infants have with their caregivers (see Ferjan Ramírez [27] for a review). Fathers have been found in some studies to talk to their children for shorter durations, and less frequently, compared to mothers [28,29], on average. One study found that infants between 3 and 20 months of age heard twice or three times as much speech from female caregivers compared to males [30]. Fathers have also been found (at a group level) to vary their pitch to a lesser degree compared to mothers when speaking to infants [14,29,31], while still varying their pitch more in IDS than in ADS [32].

Moreover, infant gender has also been found in at least one study to affect the IDS infants experience. Johnson et al. found that mothers preferentially responded to infant females whereas fathers preferentially responded to infant males [33].

Although this body of work provides evidence that gender plays some role in shaping IDS and ADS, it is important to note that the research focuses on a narrow range of family structure types, underrepresents gender nonconforming individuals, and often overlooks individual differences as well as interactions between contextual factors and demographics of the speaker as well as of the listener. Vocal communication could potentially be subject to multiple interactions between these and other factors.

## Language input in children's everyday environments

Research on early childhood development is increasingly using instruments and methodologies that capture real-world inputs, including infants' language environments, from the child's perspective. For instance, LENA's infant-worn audio recorders can capture daylong recordings of the infant's audio environment [34]. At this point, multiple repositories make widely available audio and even video recordings that capture infants' language exposure, vocal productions, and interactions with objects and others in their home environment [35,36]. These data can provide rich insights into infants' real-world experiences that cannot be captured in a lab setting.

Some research based on such recordings has highlighted the diversity in infants' everyday experiences [37]. For example, the amount of speech infants hear varies by the hour and by the speaker [38]. Variability is also found in the faces [39] and even music [40] that infants experience across a given day. Other research has highlighted consistencies such as that infants experience specific words and objects within consistent locations and events (e.g., toothbrush in the bathroom [41,42]).

Particularly relevant for the current study, researchers have used day-long infant-centered audio recordings to analyze pitch contours of real-world adult speech heard by infants over the course of a day. Both similarities and differences have been found in comparison to studies of adult speech samples elicited in a lab setting. Pitch has been found to be reliably higher for IDS versus ADS in these real-world data, similar to elicited samples, but pitch variability and predictability effects have been less robustly replicated [43,44].

## Current study

The current study extends the study of IDS and ADS acoustics within daylong audio recordings, exploring the extent to which findings from prior studies using more restricted sampling contexts translate to real-world settings and the ways in which pragmatic context, adult gender, and infant gender may modulate the features of the speech infants experience in their home environments. We leverage an existing cross-sectional, multi-site North American dataset of real-world IDS and ADS samples [30]. Specifically, we aim to (1) estimate the frequencies with which the infants experienced different

pragmatic contexts of adult speech, separating those experiences out by whether the speech they experienced was in the IDS or ADS register, (2) assess how pragmatic contexts relate to mean pitch and pitch variability, and to IDS-ADS differences in these measures, and (3) explore how adult speaker gender and infant gender are associated with the patterns observed through aims 1 and 2.

Note that we use an automatic estimate of fundamental frequency on a log scale—$\log(f_0)$—as a proxy for pitch (see Methods for details about the algorithm used for $f_0$ estimation). Pitch refers to how $f_0$ is perceived by a listener. Log-scaling $f_0$ in Hertz (Hz) accounts for the tendency for a multiplication of $f_0$ by a fixed factor to be perceived as a fixed amount of change in pitch. For example, doubling $f_0$ is perceived by listeners as an octave change in pitch, regardless of whether the $f_0$ being doubled is low in Hz or high in Hz. In contrast, an equal difference in raw Hz would be heard as relatively large or small depending on whether $f_0$ is low or high, respectively. Log-scaling $f_0$ in Hz accomplishes similar approximations to semitone, Mel, Bark and equivalent rectangular bandwidth (ERB) scales, although there are minor differences among the latter three and a log scale is the simplest and has been argued to be the optimal scale on which to measure approximate pitch differences [45].

Additionally, we use the term "gender" in this paper because pitch differences in vocal properties and the use of IDS and ADS are likely influenced by social constructs and cultural norms as well as by physiological factors. We recognize that infants in this study have not yet developed a gender identity and that their reported gender in this study reflects how they are being socialized. In addition, it should be noted that adult speaker gender in this study is based on listeners' perceptions of whether a voice sounded male or female, and that this may not correspond to the actual gender identity of the adult speaker. Thus, any differences we observe related to perceived adult gender can be expected to relate primarily to trends for gender-typical voice differences. In other words, any findings with respect to gender categories in this dataset should be interpreted as reflecting gender norms more than actual gender differences.

## Method

### Dataset

This study used the IDSLabel dataset [30], from the HomeBank repository [36]. It consisted of sound clips from four corpora of daylong LENA recordings, each collected and contributed by a separate research group between 2012 and 2016 [46–49]. Participants provided written consent to participate in the study and to have their recordings shared; this consent was obtained by the contributors of each HomeBank corpus and data collection and sharing protocols were approved by those contributors' IRBs. Participants were 60 typically developing infants (28 female) from primarily-English-speaking North American families, with cross-sectional recordings collected between 3- and 20-months-old. Forty-three infants had parents with university education. The metadata from these corpora did not consistently include race/ethnicity demographics; in some cases, recording dates are available within the original HomeBank corpora metadata. The current study's secondary analyses were approved by the University of California, Los Angeles Institutional Review Board (#21–000478) and the need for consent was waived. Data was accessed on April 13th, 2021. The authors did have access via HomeBank information that could potentially identify individual participants, including the audio recordings themselves and (in some cases) birthdates (except for the Warlaumont corpus, for which A.S.W. did have access to more identifying details, as the original contributor of that HomeBank corpus; none of those details were used or referenced as part of the current study).

To create the IDSLabel dataset, Bergelson et al. (2019) used the daylong audio (WAV) files together with the labels (provided by the LENA Pro software and available within each HomeBank corpus) to identify and extract probable adult speech clips. Adult speech clips varied in duration, but were always at least 1 second and could in principle include more than one adult speaker utterance if they were not separated by a substantial duration of silence or other sound types [50]. For each automatically identified adult speech clip, trained human coders verified that the clip was indeed adult speech and then classified the clip as adult-directed speech (ADS) or infant-directed speech (IDS), as well as categorizing the clips as male or female based on perceived speaker gender [30]. The IDS vs. ADS classification was based on the

perceived speech register of the clip; listeners were asked to make their decision based on whether the clip sounded like the speech and voice were in a style typical for addressing an infant or young child as opposed to another adult. In the current project, these individual adult speech clips were further tagged using a pragmatic context coding scheme developed specifically for this project, described in the next section.

Our study also utilizes the previous pitch estimates obtained by a prior study [43] that obtained estimated pitch contours for the adult vocalization clips in the IDSLabel dataset and calculated the mean and standard deviation (to measure within-utterance variability) of each pitch contour (i.e., $\log(f_0)$). That study used the soundgen R package's "analyze" function with the following parameter settings: pitchFloor = 75 Hz; pitchCeiling = 650 Hz; silence = 0.001; autocorThres = 0.7; pathfinding = "fast"; pitchMethods = "autocor", "spec", and "dom"; entropyThres = 0.6; step = 10 ms; wn = "hanning"; and windowLength = 50 [51].

Our dataset started with 3727 adult speech audio clips that had been tagged according to pragmatic context(s). We then excluded clips ($n$ = 107; 81 IDS; 26 ADS) that were exclusively tagged as "noisy", or indecipherable. We dropped a further 13 speech clips from the dataset because they did not include a tag for adult speaker gender. Our final dataset thus includes a total of 3607 adult speech clips (2210 IDS; 1397 ADS) after exclusions. Of these 3607 clips, 1125 clips were perceived as adult male speakers and 2482 were perceived as adult female speakers. Clips were in English except for one group of clips ($n$ = 9), all from the same LENA "conversational block", in which the adults spoke Spanish. A research assistant fluent in Spanish tagged pragmatic contexts for those clips.

### Annotation

**Coding scheme.** Table 1 shows the eight pragmatic context codes for tagging each adult speech clip. The coding scheme was developed after a review of research on parent-child communicative intent [14,52–57]. Codes were not mutually exclusive; clips could be annotated for more than one pragmatic context (or for none). Context codes were applied to both ADS and IDS clips. An additional *noisy* category was included when pragmatic context was not decipherable.

**Annotation procedure.** Trained annotators were assigned a randomized 5% of the IDSLabel dataset to annotate in each pass of coding. Each pass consisted of one LENA conversational block's worth of adult clips from each participant. A conversational block consisted of a sequence(s) of clips separated by silences less than 5 seconds [30]. Annotators were blind to child gender. They were not provided with the child's gender and did not listen to infant speech clips.

To establish annotator reliability, four annotators (including the first and second author) completed two training rounds in which reliability was established across coders for each context category. An additional two rounds of annotator

**Table 1. Abridged definitions of pragmatic contexts.**

| Pragmatic Context | Definition (abridged) |
| --- | --- |
| Conversational Basics | Standard social niceties found in speech |
| Comfort | Soothing another adult or child |
| Singing | Singing or humming a song or tune |
| Inform | Explicit, detailed information conveyed |
| Reading | Reading written material |
| Imperative | Requests and commands for another person to do something |
| Question | Question(s) posed |
| Vocal Play | Pre-linguistic sounds (i.e., babbling or cooing) |

*Note.* See the coding protocol on the project's OSF page for full definitions. Contexts were non-mutually exclusive, meaning one clip could be annotated for more than one context.

reliability were completed between the first and second author before annotating pilot data for a total of four reliability rounds reported below. Inter-coder reliability was analyzed using Krippendorff's alpha with 0 being no agreement and 1 being complete agreement [58]. The following reported mean alpha values for each context category was calculated by first averaging alpha values across pairs of coders for each context category in each round. Then, we averaged those values across the four rounds to establish an average reliability value for each category (in parentheses are each category's range of scores across all pairs of coders and across the four rounds of training): conversational basics ($M_{alpha}$=0.55; Range=0.30−.92), comfort ($M_{alpha}$=0.92; Range=0.66–1.00), singing ($M_{alpha}$=0.99; Range=0.94–1.00), inform ($M_{alpha}$=0.70; Range=0.38–0.94), reading ($M_{alpha}$=1.00), imperative ($M_{alpha}$=0.72; Range=0.21–0.97), question ($M_{alpha}$=0.77; Range=0.65–0.97), vocal play ($M_{alpha}$=1.00). Note that reading and vocal play were relatively rare categories that had perfect intercoder agreement because none of the coders identified any instances of those categories within the subset of clips for which reliability was assessed. It may also be observed that the conversational basics category had the lowest reliability average compared to other categories. This might be related to the diversity of types of functions included among conversational basics: greetings, backchanneling, polite phrases, exclamations, etc. Note also that some of the reliability coding was performed before revisions to the coding scheme; however, all codes used for the analyses reported below were made using the final version of the coding scheme as recoding was done for files coded under earlier versions. In cases where multiple coders coded a block for reliability purposes, discrepancies were resolved through discussion.

### Preregistration and pilot study

A pilot study was conducted with 505 audio clips (314 IDS and 191 ADS clips) annotated by the first and the second author. The pilot study and data aided in determining the final study design and in preparing a study preregistration. The final dataset included four coding passes plus the pilot data and training round annotations. The preregistration, coding scheme, and pilot results can be found on OSF (https://osf.io/va7c8/?view_only=63b948f200654b9ab767a4cf3e351c59).

**Preregistered planned analyses and pilot analyses.** This paper includes two sets of planned analyses: (1) prevalence of pragmatic contexts as a function of register and (2) pitch characteristics in specific pragmatic contexts as a function of register and speaker gender. In the preregistration, we planned to analyze the prevalence of pragmatic contexts as a function of register and speaker gender using logistic mixed effects models. We decided to deviate from that prior design, instead adopting a Bayesian approach to model the prevalence of pragmatic contexts as a function of register, due to sparseness of the data in some contexts. We also excluded speaker gender in order to obtain model convergence. For the preregistration, pilot analysis of adult speaker gender and register on pitch characteristics in specific pragmatic contexts utilized a separate linear mixed effects model for each pragmatic context that met our a priori analysis criterion of at least 20 clips in both IDS and ADS registers. This criterion was somewhat arbitrary and was based on our intuitions about what would be a reasonable minimum sample size for clip-level acoustic analyses; our main aim in specifying this criterion ahead of time was to avoid data dredging. We did not examine interaction terms in our models as we do in the current study due to power concerns with the pilot dataset. Another difference to note is that the variables in the pilot analysis were not centered and scaled as they are in the current study.

**Exploratory analyses.** Two of the analyses reported in the current study were not included in our preregistration. First, the analysis of pitch characteristics predicted by adult speaker gender and pragmatic context within IDS clips only was an approach we developed to better isolate the roles of speaker gender and pragmatic context, and their potential interaction, while keeping register constant, focusing on IDS as the primary register of this study. Second, the joint analysis of both adult speaker gender and infant gender on pitch characteristics within specific pragmatic contexts was not something we originally envisioned at the time of preregistration; our interest in exploring the potential influence of infant gender emerged later in the project.

## Statistical analyses

Analyses comparing how prevalent each context was in IDS versus ADS and were run in R version 4.4.1 [59] using the *brms* (2.22.0) package; a Bayesian approach was used due to the small sample sizes for some context and register combinations [60]. Analyses to compare pitch measures across contexts, registers, and gender groups were run in R version 4.3.1 using the *lme4* (1.1−34; [61]) and *lmerTest* (3.1−3; [62]) packages with restricted maximum likelihood (REML) as the estimation method. Our predictors were dummy coded: register (ADS=0; IDS=1), adult speaker gender (female=0; male=1), infant gender (female=0; male=1), and pragmatic context (non-presence=0; presence=1). Pitch measurements were log transformed, which is important to control for the nonlinear nature of pitch perception (and production), and then scaled by one standard deviation and centered using the *scale* function in R. We did not scale and center the register variable before entering it into the Bayesian mixed effects model as the variable being predicted, however, it was scaled and centered for inclusion as a predictor in the acoustic analyses, as were adult gender, infant gender, and pragmatic context. Subject identity and coder were entered as random intercepts for each model, based on likelihood ratio tests using *varCompTest* from the *varTestnlme* R package (1.3.5; [63]). Post-hoc pairwise comparisons were run using the R packages *emmeans* (1.8.7; [64]) applying Holm-Bonferroni corrections to account for multiple comparisons. Confidence intervals reported below in brackets next to the coefficients are 95% confidence intervals. Post hoc follow ups to significant interactions were interpreted by comparing the confidence intervals of the regression slopes and examining which of those post hoc regressions were statistically significant. All visualizations were created using the R package *ggplot2* (3.4.2; [65]).

## Results

### Prevalence of pragmatic contexts

Table 2 shows how many clips in the dataset were tagged as belonging to each combination of register, adult speaker gender, and pragmatic context, as well as the percentage of clips with each context, separately for each register and adult speaker gender combination. Ninety clips did not receive any pragmatic context code, and 838 clips received more than one code. We first examined the relative frequencies of each pragmatic context in IDS versus ADS. We ran a Bayesian mixed effects model with register as the outcome variable (assuming a Bernoulli distribution), each pragmatic context as a fixed effect, and participant ID and coder ID as random effects. We initially attempted to include perceived adult speaker gender and its interactions with context as an additional fixed effect, but we removed adult speaker gender in order to achieve model convergence. Our model was as follows:

$$register \sim conversational\ basics\ +\ comfort\ +\ singing\ +\ inform\ +\ imperative$$
$$+\ question\ +\ reading\ +\ vocal\ play\ +\ (1|coder)\ +\ (1|participant).$$

We ran the model with 10 chains of 10,000 iterations, adapt_delta=0.9 and max_treedepth=12, resulting in all rhats=1.0, bulk ESSs ranging from 6,844–52,284 and tail ESSs ranging from 5,250–37,140.

The results are shown in Table 2. Conversational basics and inform contexts were more prevalent in the ADS register compared to the IDS register. Questions, Imperatives, Reading, Singing, Comfort, and Vocal Play were more frequent in the IDS register compared to the ADS register.

### Acoustic analyses

We next examined two key utterance-level acoustic characteristics, mean pitch and pitch variability (i.e., standard deviation of the pitch contour), in three different analyses.

**Perceived adult speaker gender and register analysis within specific pragmatic contexts.** Our first acoustic analysis examined pitch and pitch variability of clips as a function of register and perceived adult speaker gender, separately for each of the four contexts that met our a priori analysis criterion of at least 20 instances in both

registers: conversational basics, inform, questions, and imperative (Fig 1). For each analysis, the data were first subsetted to only include clips in which the context in question was present. This set of analyses addressed the question of whether prior findings of higher overall pitch and greater pitch variability in IDS versus ADS apply in naturalistic utterances belonging to each of those contexts, controlling for speaker gender (which is strongly associated with pitch differences) and allowing detection of possible interactions between speaker gender and register. The basic model structure run for each model is as below:

$$Log\ pitch\ characteristic\ \sim\ adult\ speaker\ gender * register\ +\ (1 \mid coder)\ +\ (1 \mid participant)$$

We chose to run separate models for each context because we were interested in examining the effects of adult speaker gender and register on pitch characteristic differences within specific pragmatic contexts. See S1 File for results using alternative analysis approaches that prioritize different research questions.

**Inform context.** 2040 clips from 60 participants were tagged as inform. For mean pitch within these clips, we found significant main effects of adult speaker gender and register (Table 3); we also found an interaction for register and adult speaker gender such that the IDS-ADS difference was greater for female speakers than for male speakers and the male-female difference was greater for IDS than for ADS. For pitch variability within inform clips, a significant main effect of register was found (β = 0.14 [0.09, 0.19], $SE$ = 0.02, $p$ < .001) meaning that inform clips were significantly more variable in IDS than in ADS. All other effects were non-significant.

**Conversational basics context.** 685 clips from 57 participants were tagged as conversational basics. For mean pitch of these clips, we found significant main effects of adult speaker gender and register (Table 4). We also found a significant interaction between register and adult speaker gender such that the female-male difference was greater for IDS than ADS and the IDS-ADS difference was greater for females than for males. For pitch variability within conversational basics clips, a significant main effect of register was found (β = 0.08 [0.01, 0.16], $SE$ = 0.04, $p$ = .02). No other effects were significant.

**Table 2. Prevalence regression results and raw frequencies of clips categorized by context, register, and adult speaker gender.**

| Pragmatic context | IDS regression coefficient | Male | | Female | |
|---|---|---|---|---|---|
| | (95% CI) | ADS | IDS | ADS | IDS |
| Inform | −0.55 (−0.68, −0.43) | 252 (64.1%) | 248 (33.9%) | 888 (66.7%) | 652 (33.6%) |
| Conversational Basics | −0.13 (−0.22, −0.03) | 74 (18.8%) | 111 (15.2%) | 252 (18.9%) | 248 (12.8%) |
| Question | 0.21 (0.10, 0.32) | 58 (14.8%) | 132 (18.0%) | 154 (11.6%) | 401 (20.6%) |
| Imperative | 0.55 (0.43, 0.68) | 8 (2.0%) | 80 (10.9%) | 35 (2.6%) | 269 (13.9%) |
| Reading | 15.93 (1.68, 74.01) | 0 (0%) | 63 (8.6%) | 0 (0%) | 169 (8.7%) |
| Singing | 1.08 (0.75, 1.53) | 1 (0.3%) | 65 (8.9%) | 1 (0.08%) | 159 (8.2%) |
| Comfort | 0.17 (0.00, 0.38) | 0 (0%) | 21 (2.9%) | 2 (0.15%) | 19 (1.0%) |
| Vocal Play | 37.41 (1.02, 183.97) | 0 (0%) | 12 (1.6%) | 0 (0%) | 25 (1.3%) |
| Total | | 393 | 732 | 1332 | 1942 |

*Note.* Frequencies for the context categories do not add up to total number of clips (3607) because a non-mutually exclusive coding scheme was used. In parentheses are the regression 95% confidence intervals and the column-wise percentages.

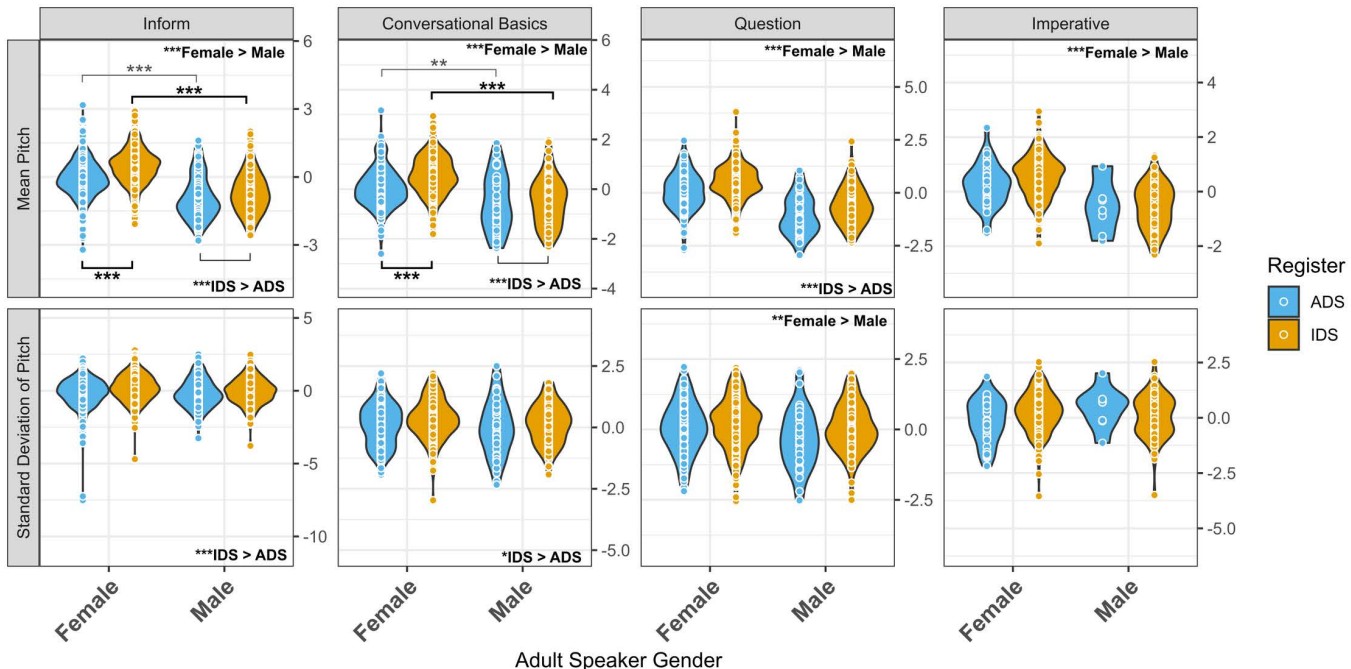

**Fig 1. Pitch and pitch variability of clips as a function of register and adult speaker gender.** *Note.* Each point within the violin plots shows the mean pitch (top row) or pitch variability (bottom row) of a clip. The four contexts for which acoustics were analyzed as a function of register and adult gender are each depicted in a separate pair of plots, one plot for each acoustic characteristic. Adult speaker gender main effects are denoted in text at the top right of each plot. Speech register main effects are denoted in text at the bottom right of each plot. Post-hoc pairwise comparisons following up on significant interactions are denoted by brackets with asterisks indicating significance level of the comparison and the bracket and asterisks bolded when a post-hoc comparison is statistically stronger than its counterpart; gender-comparison post-hocs are above the violins and register-comparison post-hocs are below the violins.

**Table 3. Effects of adult gender and register on mean pitch in the inform context.**

| Predictor | | Mean Pitch | | | |
|---|---|---|---|---|---|
| | | Estimate | SE | CI | p |
| Intercept | | 0.01 | 0.05 | −0.10–0.11 | .905 |
| Adult Gender | | −0.43 | 0.02 | −0.47 – −0.38 | **<.001** |
| Register | | 0.19 | 0.02 | 0.15–0.23 | **<.001** |
| Adult Gender*Register | | −0.09 | 0.02 | −0.13 – −0.05 | **<.001** |
| Female-Male | ADS | 0.72 | 0.07 | 0.57–0.86 | **<.001** |
| Female-Male | IDS | 1.13 | 0.08 | 0.96–1.30 | **<.001** |
| IDS-ADS | Female | 0.49 | 0.05 | 0.38–0.61 | **<.001** |
| IDS-ADS | Male | 0.08 | 0.08 | −0.11–0.26 | .36 |

*Note.* Linear mixed effects model results. The four rows below the interaction term row give the post-hoc analysis results. The direction of effect for post-hoc findings is such that the category before the minus sign is greater than the category following the minus sign if the effect is positive.

**Question context.** 745 clips from 60 participants were tagged as question. For mean pitch of these clips, we found significant main effects for both adult speaker gender (β = −0.48 [−0.55, −0.41], *SE* = 0.04, *p* < .001) and register (β = 0.19 [0.12, 0.26], *SE* = 0.04, *p* < .001). For pitch variability within question clips, we found a significant main effect of adult speaker gender (β = −0.11 [−0.18, −0.03], *SE* = 0.04, *p* = .006). No other effects were statistically significant.

*Imperative context.* 392 clips from 51 participants were tagged as imperative. For mean pitch of these clips, a significant main effect for adult speaker gender was found (β = −0.43 [−0.57, −0.30], *SE* = 0.07, *p* < .001). No other significant effects were identified.

**Comparison of adult speaker gender and pragmatic context within IDS.** Our second set of acoustic analyses addressed whether there were differences in mean pitch and pitch variability of IDS clips across contexts, considering perceived speaker gender (*n* = 2210 IDS clips, 60 participants; Fig 2). Vocal play was not included due to there being fewer than 20 IDS vocal play clips. Complete regression tables and interaction plots are in S1 File.

*Mean pitch within IDS.* The linear mixed effect model to test for effects of adult gender and pragmatic context on mean pitch of IDS clips was formulated as:

$$Log\ mean\ pitch\ \sim\ adult\ speaker\ gender * conversational\ basics\ +\ adult\ speaker\ gender * comfort$$
$$+\ adult\ speaker\ gender * singing\ +\ adult\ speaker\ gender * inform\ +\ adult\ speaker\ gender * imperative$$
$$+\ adult\ speaker\ gender * question\ +\ adult\ speaker\ gender * reading\ +\ (1|coder)\ +\ (1|participant)$$

We found a significant main effect of adult speaker gender (β = −0.47 [−0.52, −0.42], *SE* = 0.03, *p* < .001) and a significant interaction between adult speaker gender and singing (β = −0.05 [−0.08, −0.01], *SE* = 0.02, *p* = .007). Post hoc pairwise comparisons revealed that adult male speakers used significantly lower pitch in singing versus non-singing IDS contexts (β = −0.39 [−0.71, −0.07], *SE* = 0.14, *p* = .01), and that this difference was significantly greater for adult males than for adult females, who did not exhibit a significant difference in mean pitch between singing and non-singing IDS.

*Pitch variability within IDS.* The linear mixed effect model for pitch variability within IDS was:

$$Log\ pitch\ variability\ \sim\ adult\ speaker\ gender * conversational\ basics\ +\ adult\ speaker\ gender * comfort$$
$$+\ adult\ speaker\ gender * singing\ +\ adult\ speaker\ gender * inform\ +\ adult\ speaker\ gender * imperative$$
$$+\ adult\ speaker\ gender * question\ +\ adult\ speaker\ gender * reading\ +\ (1|coder)\ +\ (1|participant)$$

We found a significant main effect of adult speaker gender (β = −0.09 [−0.14, −0.04], *SE* = 0.03, *p* = .001) such that females varied their pitch more than males. We also found main effects of singing (β = −0.19 [−0.23, −0.15], *SE* = 0.02, *p* < .001), conversational basics (β = 0.05 [0.004, 0.09], *SE* = 0.02, *p* = .03), and comfort (β = 0.06 [0.02, 0.09], *SE* = 0.02, *p* = .004). Singing clips were less variable and conversational basics and comfort clips were more variable than other contexts. We also found a significant interaction between adult speaker gender and singing (β = −0.06 [−0.09, −0.02],

**Table 4. Effects of adult gender and register on mean pitch in the conversational basics context.**

| Predictor | | Mean Pitch | | | |
|---|---|---|---|---|---|
| | | Estimate | SE | CI | p |
| Intercept | | 0.10 | 0.08 | −0.06–0.27 | .206 |
| Adult Gender | | −0.33 | 0.04 | −0.40 − −0.25 | **<.001** |
| Register | | 0.16 | 0.04 | 0.09–0.23 | **<.001** |
| Adult Gender*Register | | −0.11 | 0.03 | −0.18 − −0.04 | **.001** |
| Female-Male | ADS | 0.43 | 0.12 | 0.16–0.71 | **<.001** |
| Female-Male | IDS | 0.95 | 0.12 | 0.68–1.23 | **<.001** |
| IDS-ADS | Female | 0.45 | 0.09 | 0.26–0.65 | **<.001** |
| IDS-ADS | Male | −0.06 | 0.14 | −0.37–0.25 | .68 |

*Note.* Linear mixed effects model results. The four rows below the interaction term row give the post-hoc analysis results. If the effect is positive, then the category before the minus sign tended to have greater values than the category after the minus sign.

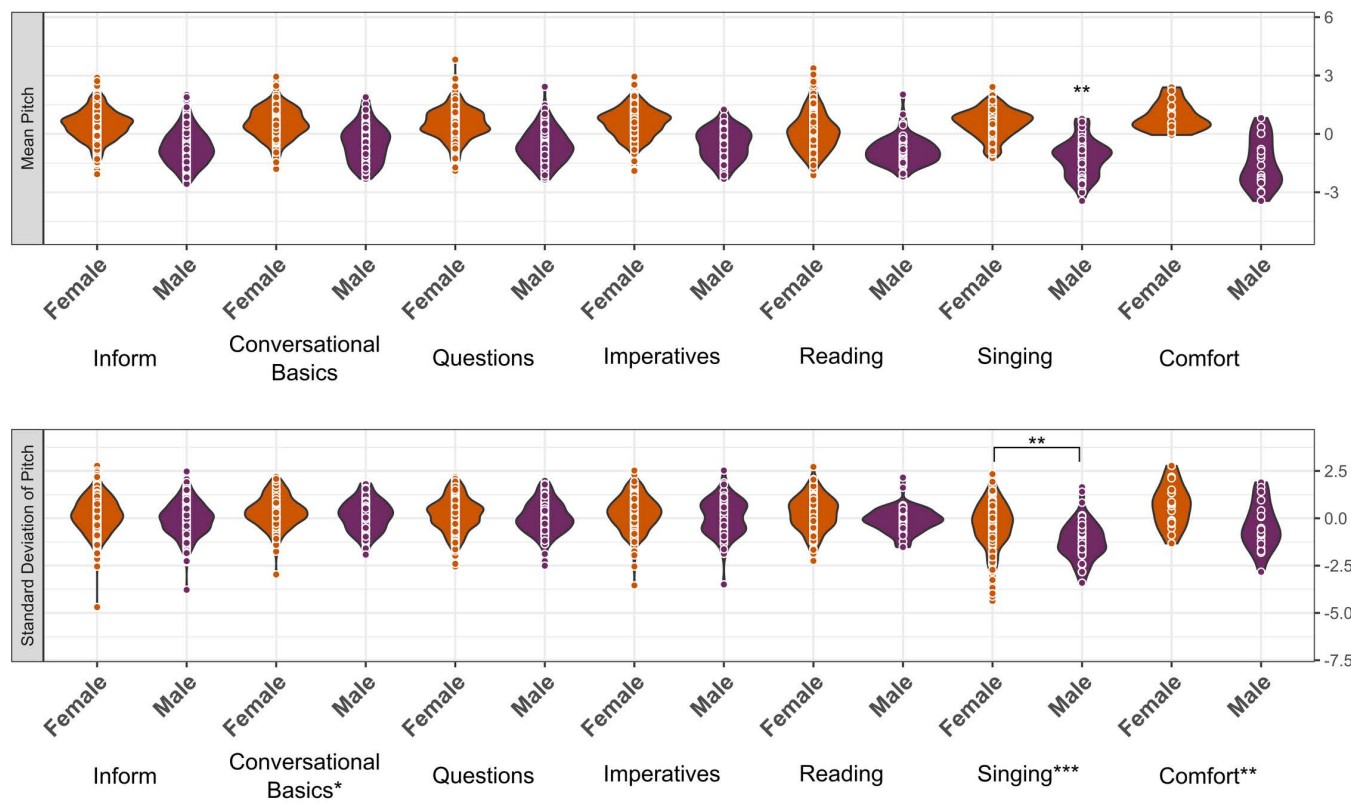

**Fig 2. Mean Pitch and Pitch Variability of IDS Clips as a Function of Adult Speaker Gender and Pragmatic Contexts.** *Note.* Each point within the violin plots shows the mean pitch (top row) or pitch variability (bottom row) of a clip. Only IDS clips are represented in this figure and in the corresponding analyses (see text). Asterisks next to pragmatic context categories denote significance of context main effect (the singing main effect was negative, and the conversational basics and comfort main effects were positive). Asterisks above the violins denote significance of the post hoc following up on speaker gender * context interactions.

$SE = 0.02$, $p = .005$). Post hoc pairwise follow-up comparisons revealed that the difference in pitch variability between singing and non-singing contexts was significantly greater for adult males than for adult females and that the female-male difference in pitch variability was greater for singing than for non-singing contexts.

**Adult speaker gender and infant gender within specific IDS contexts.** Our third set of acoustic analyses focused on mean pitch and pitch variability of IDS clips as a function of infant gender and perceived adult speaker gender, using separate models for each pragmatic context. These analyses tested whether the gender of the infant likely being addressed has any effects on speaker pitch, controlling for perceived adult speaker gender and enabling detection of possible interactions between infant gender and adult gender. We chose to analyze contexts separately rather than to attempt to include context as a fixed effect to increase the interpretability of the results. See S1 File for results of a single model with pragmatic context included as predictor variables. Vocal play was again excluded due to having less than 20 instances. The basic model structure run for each model was:

$$Log\ pitch\ characteristic\ \sim\ infant\ gender * adult\ speaker\ gender\ +\ (1\,|\,coder)\ +\ (1\,|\,participant)$$

In the conversational basics, reading, question, imperative, inform, and singing models, we found a significant main effect of perceived adult speaker gender in that females used a higher mean pitch when speaking to infants than males. Singing was the only context for which there was a significant effect for pitch variability. No significant effects were found in the comfort context for either mean pitch or pitch variability. None of the contexts had significant main effects for infant gender. However, we found two cases of significant interactions between infant gender and adult gender, reported below and visualized in Fig 3.

For pitch variability of inform context clips (900 IDS clips, 59 participants), there was a significant interaction between infant gender and perceived adult gender (β = −0.08 [−0.16, −0.01], SE = 0.04, p = .03). Post hoc pairwise comparisons revealed that male infants experienced a greater difference between adult females' and adult males' pitch variability, with adult females' pitch variability being significantly higher than adult males' for male infants (β = 0.41 [0.13, 0.68], SE = 0.12, p = .002).

For mean pitch of singing clips (224 IDS clips, 24 participants), there was a significant interaction between perceived adult speaker gender and infant gender for mean pitch (β = 0.23, [0.08, 0.37], SE = 0.07, p = .002). Post hoc pairwise revealed that for female infants, there was a greater difference between adult male and adult female mean pitch. Moreover, adult males used significantly higher mean pitch with male infants than female infants (β = −0.99 [−1.72, −0.27],

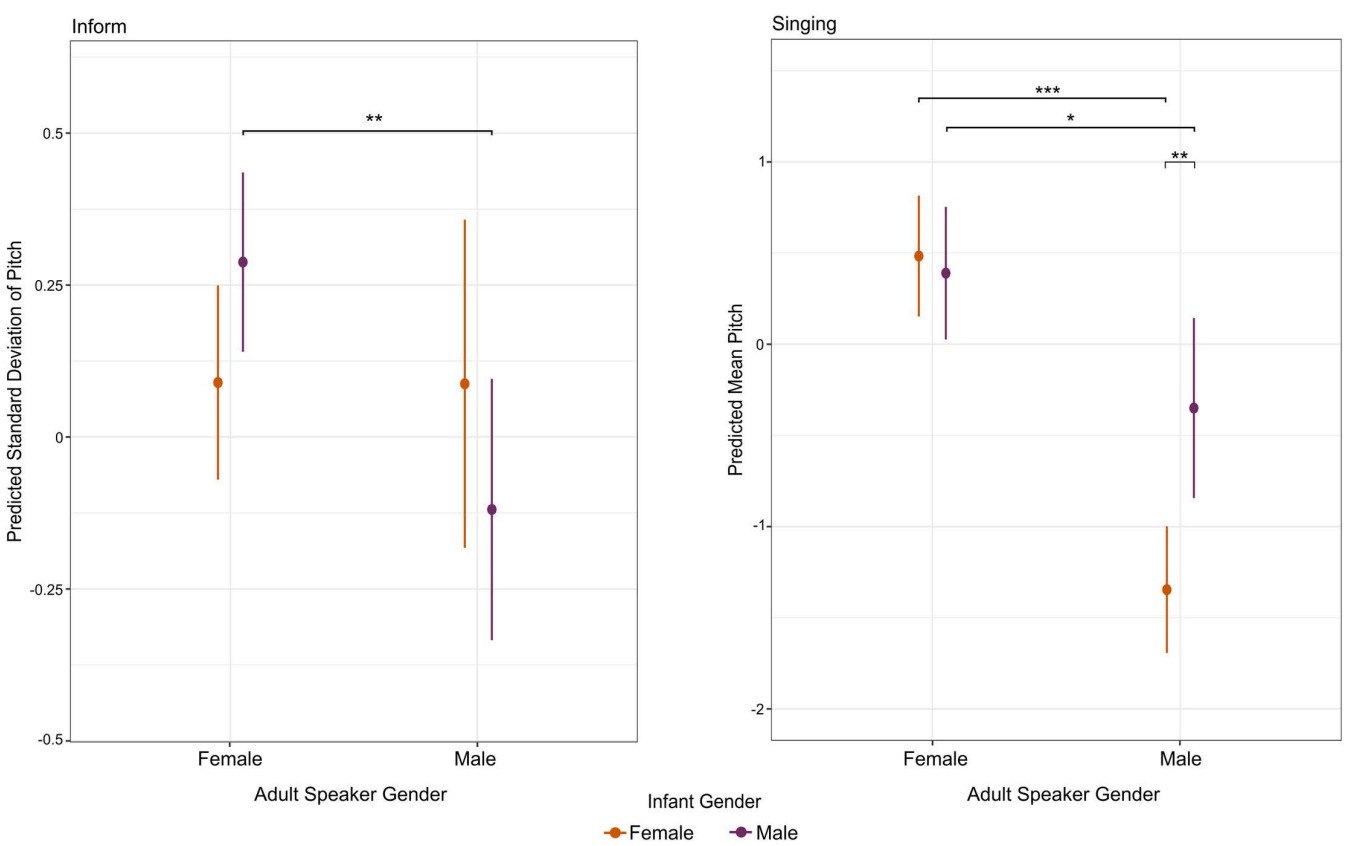

**Fig 3. Estimated Marginal Means of Adult Speaker Gender and Infant Gender Interactions in the Inform and Singing Contexts.** *Note.* Estimated marginal means of the post hoc follow up tests for the significant interaction between adult speaker gender and infant gender effects on pitch variability in the inform context and on mean pitch in the singing context. Error bars denote 95% confidence intervals and *'s denote which post hoc follow up tests were statistically significant.

*SE* = 0.31, *p* = .006), a trend that was significantly different from that of adult females, for whom there was no significant effect of infant gender on mean infant-directed singing pitch.

## Discussion

### Prevalence of pragmatic contexts

By analyzing randomly sampled real-world day-long recording data, we could get a lay of the land of how frequently this group of young North American children experienced different pragmatic types of adult speech, and whether those experiences occurred in IDS versus ADS register. Question, imperative, reading, singing, comfort, and vocal play contexts were more frequent in IDS compared to ADS, in line with prior research finding imperatives, reading, and questions to be frequent in children's everyday input [38] along with music exposure consisting of live voices, including caregivers' singing [40].

In contrast, inform and conversational basics contexts were more frequent in ADS compared to IDS. However, both these contexts were the most prevalent categories regardless of adult speaker gender and register, so infants received a fair amount of IDS input in these contexts as well. The high prevalence of inform clips supports the everyday relevance of the large body of work demonstrating for information-providing contexts that IDS is more salient and attractive to infants and that it also has features that tend to make learning language and concepts easier for infant learners [5,10,66].

As for conversational basics, relatively less literature has focused on IDS-ADS contrasts in this domain. However, there is a literature documenting that conversational basics utterances, such as salutations (e.g., "hi" and "bye"), are common early words in infants' vocabulary [67,68]. The fact that conversational basics clips are even more prevalent in adult-directed speech in these daylong recordings indicates that these types of clips are not transient elements of early childhood language experience but clip types that infants are frequently exposed to, can learn early, and will continue to utilize throughout their lifespans.

Vocal play and comfort were the least frequent clip types overall. The low prevalence of comfort surprised us, given that, anecdotally, comforting or soothing constitutes much of what caregivers seem to do over the course of the day. The finding may relate to the way in which the category was defined; it is possible that much comforting takes place without explicit language that would have cued our listeners to make a confident judgment that an clip was a comfort clip [14,69]. Another possibility is that comforting in everyday contexts often takes place without accompanying adult speech. Also, anecdotal impressions of the frequency of a vocal pragmatic type do not necessarily correspond reliably to its actual frequency but may instead relate more closely to the experiential salience of the behavior—this mismatch has been proposed to explain the discrepancy between perceived prevalence of infant cries and laughs and their actual low frequencies of occurrence in day-long recordings compared to pre-speech sounds [70]. We also expected that vocal play might be higher, given the literature on adult imitation of infant babble [71]. As with comfort clips, it is possible that that the operationalization of vocal play in the current study was not as inclusive as in prior studies, or this behavior may be rarer than prior literature and anecdotal experience would suggest. Furthermore, both categories could have been interpreted as noise and coded exclusively as "noisy" clips and consequently excluded from our final dataset, and comfort could have occurred overlapping child crying and tagged as such by LENA, in which case it would not have been included among the clips in IDSLabel.

Finally, it is possible that the prevalence of the various register-context combinations could differ as a function of perceived adult gender and/or assigned child gender. The current study did not have sufficient sample size to analyze the prevalence results at that level of granularity.

### Acoustic analyses

**Perceived adult speaker gender and register within specific pragmatic contexts.** In the inform, conversational basics, and question categories, we found IDS clips to have significantly higher mean pitch than ADS clips. This finding provides real-word validation of higher pitch in IDS compared to ADS for all three of these contexts. However, we also

found that for the inform and conversational basics contexts there was a significant interaction between speaker gender and register, such that the IDS-ADS pitch difference was more pronounced for speakers perceived as female than for speakers perceived as male. We also found that for the inform and conversational basics contexts, IDS clips were more variable in pitch than ADS clips which is consistent with prior literature [3,43].

**Perceived adult speaker gender and comparisons between pragmatic contexts within IDS.** When looking across IDS clips from different pragmatic contexts, we found that comfort IDS involved greater pitch variability than non-comfort IDS and that infant-directed singing involved less pitch modulation than non-singing IDS. However, for both mean pitch and pitch variability, we also found significant interactions between perceived adult speaker gender and whether a clip involved singing, with adult speakers perceived as male exhibiting a greater reduction in mean pitch and pitch variability during singing than adult speakers perceived as female.

There are a variety of possible explanations for these findings. It is possible that parents in our sample sang mostly lullaby-type songs, which have been found to be lower in pitch and less variable than play songs [18]. The perceived adult gender effects could be related to differences in the songs that are sung, and song selection could vary by gender due to factors such as cultural, social, and individual preferences. It is also possible that perceived adult gender effects could be related to the manner in which a song is sung to infants by male versus female caregivers. In other words, these results could be influenced by biological differences in pitch range, differences in song choice, and/or differences in singing style. Alternatively, the perceived adult gender effects could reflect tendencies of the listeners who coded adult gender to be biased toward perceiving voices as male when pitch and pitch variability were lower.

The finding that comfort clips in IDS had greater pitch variability than IDS non-comfort clips was initially surprising to us because we had originally expected that comfort clips would have pitch properties with mellowing acoustic effects [72]. The finding of greater pitch variability for comfort clips might reflect the way we operationalized comfort in our listener coding protocol and/or the representation of different subtypes of comfort utterances within the dataset.

**Perceived adult speaker gender and infant gender in IDS.** Our final set of acoustic analyses explored the effects of assigned infant gender and perceived adult speaker gender on pitch and pitch variability of IDS clips within specific pragmatic contexts. In the inform and singing contexts, we found some intriguing interactions. In the inform context, female-sounding adult speakers varied their pitch more with male infants compared to female infants. In the singing context, male-sounding adult speakers used lower mean pitch with female infants compared to male infants. Apparently, adult speakers sometimes modified their pitch more when addressing male infants compared to female infants, with these infant addressee gender effects depending on perceived adult speaker gender.

These patterns might be related to differences in the types of sounds made by male versus female infants. A prior study found that female adults preferentially responded to infant males with this preferential response appearing to be related to infant males having less of a nasal quality to their vocalizations, a trait that is rated as more socially favorable [73]. Furthermore, infant males have been found to be on average more active [74] and to vocalize more [75] than infant females, which could contribute to differing vocal responses from adults. It is important to emphasize that, while the present results support the idea that infant gender may influence adult speech acoustics, the results are exploratory. Future research using other datasets and/or experimental designs should test for associations between infant gender on parents' acoustic characteristics to determine the reliability of these effects as well as gain insight into their possible explanations.

## Overall implications

Taken together, our results corroborate prior research on overall pitch and pitch variability differences between ADS and IDS, but they also highlight the relevance of various pragmatic and demographic factors and the rich interplay between those factors when it comes to acoustic characteristics of the adult speech young children are exposed to in their everyday language environments. This is significant because it is well established in the IDS literature that prosodic features of adult speech play a role in language learning. For example, infants seem to prefer to listen to IDS over ADS [76–78].

Infants' preference for IDS, possibly due to its saliency and frequency in infants' daily language input, may be related to its positive association with children's language development [5,14]. The current study brings to light some of the various factors (i.e., adult speaker gender presentation, assigned infant gender, and pragmatic context) that can affect the prosody of IDS and ADS in infants' everyday environments. This provides more fine-grained information that can be used to inform future experimental, computational modeling, and descriptive research.

Additionally, researchers could use our pragmatic context labels, combined with the original IDSLabel dataset, to ask additional questions regarding the effects of pragmatic contexts, perceived adult speaker gender, assigned infant gender, and/or register on adults' speech inputs to children. For example, the inform context includes diverse clips like labeling statements and explanations. These clips could serve different functions of speech and could be further coded and analyzed to associate those different functions with differences in prosody, word choice, or other features of the clips.

The current study supports the value of research on various factors (beyond the pragmatic and gender-related features studied here) that may complexly affect acoustic properties of adult speech in young children's everyday environments. Future research taking a more fine-grained approach to considering individual differences and demographic and situational variables (e.g., time of day or month, activity context, SES, and child age) may reveal yet more variation and context-sensitivity in prosodic differences in IDS and ADS in ecologically valid settings. For example, we did not include infant age as a factor in the current study, but prior research has shown that differences in adult speakers' mean fundamental frequency are not as strong between IDS and ADS as children age [79], and it is conceivable that this age trend in mean $f_0$ might differ across contexts, genders, etc. Like how Fernald's (1989) cross-linguistic study of IDS laid the foundation for and prompted more fine-grained follow-up studies to extend and challenge the original work (e.g., Mazuka et al., 2015), we hope the current study will inspire find-grained, follow-up work [4,14].

## Limitations

The dataset for the current study relies on infant-worn recorders [34] and on pre-processing to identify likely adult clips. The recorder and tagging software do not always capture clear input for all adult clips, and many adult clips may have been missed entirely due to being tagged as overlapping with another sound source or being mistagged as a different sound source type. These factors should also be considered when interpreting results, as should the noisiness of the audio in some cases due to using infant-worn recorders in real-world environments. However, the natural recordings used in the current study are important to establish high ecological validity, and some of that noisiness reflects noise that infants also experience (and must grapple with).

Furthermore, the current study relied on two existing datasets: the IDSLabel dataset [30] and pitch estimations from MacDonald et al. [43]. In the IDSLabel dataset, the IDS versus ADS distinction was based on the way clips sounded, rather than who the listener thought the speaker intended to address. This subjectivity in identifying register should be considered when interpretating the results of the current study. Additionally, speaker gender was based on the listener's best guess with binary gender options rather than on speakers' actual gender identities. Race, age, gender typicality, and other factors that can influence voice pitch and may further modulate how IDS and ADS vary. Future research is needed, ideally with datasets for which the speakers' multiple social identity dimensions are self-reported [80].

The current study also uses pitch estimations calculated through an automated process. While we do not have any reason to think that this process systematically biased the results, it is possible that errors in that estimation process could have affected some of our results. In the future, it would be helpful to validate the automatically obtained pitch estimates against pitch measurements obtained by expert humans. Furthermore, the current study focused on just two acoustic features. While these are the most studied features when it comes to studies of IDS, there are many other acoustic features (e.g., pitch predictability, pitch intensity, voice quality, vowel formants) which could in principle be measured and assessed in relation to the pragmatic and gender variables analyzed here [3,10,43,81].

There were also limitations introduced by the choices we made in designing the pragmatic contexts coding protocol, as noted earlier in the Methods and Discussion. In many cases, it would be helpful for interpretation if the context categories were further broken down into subcategories. This is a concern that also applies to much prior research that describes communication behavior as a function of pragmatic context or function (e.g., [3,14]). The extent to which subtypes of pragmatic contexts are associated with acoustic differences in naturalistic input, and how this might interact with demographic and other factors could be addressed in future research.

Additionally, this dataset is limited to a North American sample from just four data collection sites, limiting the generalizability of our findings. Prior work has found that IDS, specifically, is a robust phenomenon in many cultures (e.g., [1–3,82]). However, studies have also found that communicative styles in IDS differ by culture and that prosodic characteristics of IDS may not be completely universal [3,4,82]. The current study differs from the approach of the aforementioned studies in that we examined the effect of communicative intent (i.e., pragmatic contexts), gender, and register on acoustic properties in real-world daylong audio recordings. Future research could expand on the current approach with speech samples from various languages and cultures.

There is also within-culture individual variation in infants' everyday experiences [37] not explored in the current study. For example, the infants in our study range from 3- to 20- months of age. Infants' experiences in their day-to-day environment change significantly with age due to motor ability as one example [83]. Additionally, infants have different routines (e.g., [38,84]), interactions with different adults [38], and overall sensory input (e.g., faces, music, etc. [39,40]) that can affect the speech input that they hear in a given day. Taken together, these factors contribute to the amount and type of input an individual infant receives in their everyday environment. Future research could examine prosodic features of speech in infants' day-to-day environments with focus on individual differences.

Finally, examining child engagement and interactive effects related to child speech and conversation was beyond the scope of the current study and we did not analyze child vocalizations. However, prior work has found that mother-infant dyads had similar pitch characteristics in conversation with one another [85]; research on how the patterns of IDS pitch modifications found here relate to infant behavior, including infant vocalization acoustics, could shed light on both the mechanisms underlying the current study's findings and its implications for infant behavior and development.

## Conclusion

We utilized existing clips from real-world audio recordings and perceived speaker gender labels, perceived IDS register labels, and acoustic measurements for those clips, then deployed human listeners to code perceived pragmatic context. This enabled us to test whether findings from studies on IDS and ADS in more restricted contexts generalize to real-world data and to explore whether there are variations in these patterns as a function of perceived adult gender, assigned infant gender, and pragmatic context. Indeed, we found several cases of interactions between these contextual factors highlighting the rich interplay of factors that relate to infants' real-world linguistic experiences.

## Supporting information

**S1 File.** Supplementary Material.
(DOCX)

## Acknowledgments

The authors would like to thank the families who participated in the studies that are included in the IDSLabel Dataset; the other researchers who contributed the HomeBank and IDSLabel data; Kyle MacDonald, whose pitch measurements were used for this study and for discussions that inspired the current study; and Giselle Littleton and Amy Wong for their help with annotation. Special thanks to Catherine Sandhofer, the UCLA Emergence of Communication Lab members, and

the UCLA Language and Cognitive Development Lab members for helpful comments and suggestions on the project and manuscript.

## Author contributions

**Conceptualization:** Emily Neer, Anvi Brahmbhatt, Anne Warlaumont.

**Formal analysis:** Emily Neer, Anvi Brahmbhatt, Catherine Walsh, Anne Warlaumont.

**Methodology:** Emily Neer, Anne Warlaumont.

**Project administration:** Emily Neer.

**Visualization:** Emily Neer, Catherine Walsh.

**Writing – original draft:** Emily Neer, Anvi Brahmbhatt, Anne Warlaumont.

**Writing – review & editing:** Emily Neer, Anvi Brahmbhatt, Catherine Walsh, Anne Warlaumont.

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
