## [Decision Letter · Decision Letter 0]

PONE-D-24-37730Pitch characteristics of real-world infant-directed speech vary with pragmatic context, perceived adult gender, and infant genderPLOS ONE

Dear Dr. Neer,

Thank you for submitting your manuscript to PLOS ONE. After careful consideration, we feel that it has merit but does not fully meet PLOS ONE’s publication criteria as it currently stands. Therefore, we invite you to submit a revised version of the manuscript that addresses the points raised during the review process.

We look forward to receiving your revised manuscript.

Kind regards,

Marcela de Lourdes Peña Garay, Ph.D

Academic Editor

PLOS ONE

Journal Requirements:

2. Please include captions for your Supporting Information files at the end of your manuscript, and update any in-text citations to match accordingly. Please see our Supporting Information guidelines for more information: http://journals.plos.org/plosone/s/supporting-information .

Reviewers' comments:

Reviewer's Responses to Questions

**Comments to the Author**

1. Is the manuscript technically sound, and do the data support the conclusions?

Reviewer #1: Yes

Reviewer #2: Yes

Reviewer #3: Partly

2. Has the statistical analysis been performed appropriately and rigorously? 

Reviewer #1: Yes

Reviewer #2: Yes

Reviewer #3: Yes

3. Have the authors made all data underlying the findings in their manuscript fully available?

Reviewer #1: Yes

Reviewer #2: Yes

Reviewer #3: Yes

4. Is the manuscript presented in an intelligible fashion and written in standard English?

Reviewer #1: Yes

Reviewer #2: Yes

Reviewer #3: Yes

5. Review Comments to the Author

Reviewer #1: The manuscript investigates how prosodic modifications in infant-directed speech (IDS) vary based on pragmatic context, perceived adult gender, and infant gender. Analyzing 3,607 speech clips from daylong home recordings of 60 North American infants, the study assesses pitch characteristics across various speaking contexts. The authors exhibit rigor and transparency in their study design, including preregistration of their analyses. They clearly detail the method for assigning adult gender, which is noted as a study limitation—a commendable inclusion. I appreciate the authors’ distinction between planned and exploratory analyses, enhancing the clarity of the reporting. Below, I list some minor suggestions for revision:

1. The authors also provide a detailed explanation of inter-coder reliability, noting the average across four rounds of coding. Given that the practice of averaging reliability has been debated, it would be helpful if the authors also reported the range to give a fuller picture. For example, the reliability for most categories is strong, though it is lower for "Conversational Basics" (0.55). Further explanation on why this category’s reliability is comparatively low and how this may affect the interpretation of results would be valuable.

2. There are a few areas that require further clarification. For instance, it is not entirely clear if "3,727 adult speech audio clips" refers to distinct utterances. Later, the authors mention “3,607 adult speaker utterances (2,210 IDS; 1,397 ADS) after exclusions,” suggesting that all clips were treated as individual utterances. How were utterances determined? In the OSF repository, the first clip (comfort_inform_CDS) includes two grammatical segments (clauses). Were they counted as one utterance?

3. The pragmatic context coding scheme for utterances includes categories such as "Inform," "Conversational Basics," "Question," "Imperative," "Reading," "Singing," "Comfort," and "Vocal Play." It would be helpful to know whether all utterances fit into these categories or if there were cases that could not be coded within this framework.

4. Beyond the factors analyzed, I am curious if the authors considered other potential influences. For example, was the time of day when utterances occurred recorded, and could this factor influence speaker presence by gender? Additionally, given that 43 infants had parents with higher education, could parental education also play a role in prosodic modification patterns?

5. Further detail on the perceived gender distribution of adult speakers within each participant and across the total sample is recommended. Providing descriptive information about these characteristics would offer a clearer picture of the sample's demographic composition. It would also be beneficial if the authors elaborated on the automatic pitch detection methodology and whether they validated this method against human-coded data.

Overall, the manuscript offers a valuable contribution to the field, with results well-placed within the broader child development literature. While the core findings are robust and data-supported, further refinement in presentation and organization would enhance the manuscript’s clarity and impact.

Reviewer #2: First of all, I would like to congratulate the authors on this interesting study. The topic is relevant, the conducted experiment is feasible and the applied statistical analyses are well described, clear and appropriate. The authors discussed all important aspects, including the limitations of the study. I definitely recommend this manuscript for publication. I only have some minor comments and suggestions.

1. I am not familiar of LENA device and its working mechanism. On the official website I read that it’s deleting the actual conversation and store only the acoustic data in order to meet personality rights. If so, how audio playbacks were coded by the coders? The website also stated that the device is not able to “understand” spoken words but it can count the words. But it can also count utterances (as in the article there are information about the number of utterances). In the article there are also utterance-level acoustic characteristics, so it seems that the device is able to do utterance-level analyses. Is it able to do hyperarticulation analyses? Or any other vowel-level analyses? Anyway, it would be expedient to write a little bit more about the device, the software, the output, etc. for those who are not familiar with this data acquisition method.

2. It would be also nice to have a table with the number of clips in each pragmatic context category (authors noted that Reading and Vocal play were underrepresented and sometimes these contexts were excluded from analysis, but it would be important to see the exact numbers within each pragmatic context). It would be also important to see the exclusion rate of clips within each pragmatic context. For instance, we can read in the discussion that authors were surprised that comfort was rare in the dataset. But isn’t it possible that most of the comfort context had been excluded because the infant was crying and as it was considered as “noisy”? A detailed table about excluded clips would give an answer to this question.

3. Line 252 criterion of at least 20 utterances seems a little bit arbitrary. What was the basis for this inclusion criteria?

Reviewer #3: The study "Pitch characteristics of real-world infant-directed speech vary with pragmatic context, perceived adult gender, and infant gender" explores how various factors influence the acoustic properties of infant-directed speech (IDS). Using an open dataset of speech clips recorded in the naturalistic environments of North American homes with infants aged 3 to 20 months, the authors compared the pitch features of IDS with adult-directed speech (ADS). Their analysis confirmed that IDS consistently exhibits a higher mean pitch and greater pitch variability compared to ADS, aligning with previous findings. Moreover, the study revealed that these pitch characteristics are not uniform but vary depending on the pragmatic function of the utterance, the gender of the speaker, and the gender of the infant. These findings highlight the nuanced and context-sensitive nature of IDS, underscoring its potential role in facilitating infant engagement and learning.

The authors made commendable efforts to address the issue of zero counts in some pragmatic contexts by introducing a randomization-based approach to include these categories in their logistic mixed-effects model. However, the limited observations for contexts such as 'reading,' 'singing,' and 'comforting,' particularly in the ADS register, raise concerns about the robustness of the conclusions drawn for these categories. Sparse or absent data for certain pragmatic contexts inherently limit the statistical power and reliability of the model's estimates for these categories. The artificial reclassification of data to enable model convergence, while creative, introduces additional uncertainty and potential biases that complicate the interpretability of the findings for these specific contexts.

The results from running the logistic mixed-effects model provided in the manuscript's accompanying code revealed no significant main effect of the pragmatic context of questions on the register, contrary to the findings reported in the paper (see attached html table). This discrepancy reinforces concerns about the robustness of the analysis, particularly given the limited observations for some pragmatic contexts and the reliance on resampling techniques to address zero counts. Furthermore, the warning message generated during model execution—“Model is nearly unidentifiable: large eigenvalue ratio - Rescale variables?”—raises additional doubts about the model's stability and reliability.

The study analyzes IDS across a broad age range of 3 to 20 months, which spans a critical period of infant development. Previous research has demonstrated that IDS evolves as infants grow, with notable changes in pitch characteristics (e.g., Cox et al., 2023), and pragmatic functions tailored to developmental stages, such as capturing young infants’ attention and conveying emotional affect versus supporting linguistic goals in older infants (Fernald, 1992; Kuhl et al., 1997). By aggregating data across this wide age range without accounting for potential developmental differences, the study may obscure important age-specific patterns in IDS. For example, pitch characteristics or the frequency of certain pragmatic contexts (e.g., comforting vs. reading) may differ markedly between younger and older infants, which could confound the results. A more nuanced analysis that examines IDS characteristics by narrower age bands or includes age as a covariate could provide deeper insights into how pragmatic contexts and pitch modulations vary across development.

Furthermore, the inclusion of Spanish utterances in the dataset introduces an additional variable that could influence the pitch characteristics of IDS, as cross-linguistic research has shown that IDS varies significantly across languages (e.g., Fernald et al., 1989). For instance, Spanish IDS may exhibit different pitch ranges, prosodic contours, or modulation patterns compared to English IDS due to phonological and cultural differences in speech patterns (e.g., Cox et al., 2023). These linguistic variations could confound the analysis, particularly if the distribution of Spanish and English utterances is uneven across pragmatic contexts, speaker genders, or infant age groups.

Differences in the pitch characteristics of male and female speakers in the context of singing could be influenced by the type of songs they sing rather than solely reflecting their vocal pitch range or communication style. This is a critical point to consider, as song selection can vary systematically by gender due to cultural, social, or individual preferences, which may introduce confounding factors into the analysis. It is unclear whether the observed pitch differences in singing across genders reflect biological differences in vocal range, variations in song choice, or a combination of these factors.

One of the notable strengths of this study is its commitment to transparency and reproducibility, exemplified by the accessibility of both the data and analysis code. By utilizing an existing open dataset of speech recordings, the authors enable other researchers to verify their findings and extend the work in new directions. Moreover, the availability of the analysis code provides a clear roadmap for reproducing the statistical models and simulations, facilitating peer verification.

Overall, this study offers valuable insights into how pitch characteristics of IDS vary by pragmatic context, speaker gender, and infant gender, yet several limitations and potential confounds warrant careful consideration. The broad age range of 3 to 20 months introduces variability that may obscure developmental changes in IDS, as prior research has shown that IDS evolves with infants’ linguistic and social development. Additionally, the sparse or absent observations for certain pragmatic contexts, such as reading and vocal play, especially in ADS, raise questions about the robustness of the conclusions drawn from these data. The method of artificially reclassifying utterances to address zero counts, while creative, introduces uncertainty and potential bias. Lastly, the observed gender differences in the context of singing could reflect differences in song selection rather than purely biological vocal characteristics or interaction styles. Addressing these issues through more stratified age analyses, and controls for song content—would strengthen the study’s claims and contribute to a more nuanced understanding of IDS dynamics.

Minor points:

Line 115: can LENA capture 16 hours a day? Limited by battery capacity or by storage capacity?

Line 144: But how is it measured automatically?

Line 200 ff: It seems not clear to me, whether a speech clip is synonymously used for utterance and what is meant by a conversational block (especially with regard to identifying how many utterances were in Spanish).

Line 220: How were annotators bling to the child’s gender? Couldn’t this be similarly perceived as the adult speaker’s gender?

Line 330: It might be better to use identical labels within figure and manuscript – standard deviation of pitch versus pitch variability

End of tables and figures are hard to parse: Possibly add “Note” to text below tables and figures still belonging to them.

Supplemental Material Code (Alt analyses): line 122 the infant model mean is missing a + after the interaction of adu_gender*chi_gender*comfort

6. PLOS authors have the option to publish the peer review history of their article (what does this mean? ). If published, this will include your full peer review and any attached files.

**Do you want your identity to be public for this peer review?** For information about this choice, including consent withdrawal, please see our Privacy Policy .

Reviewer #1: No

Reviewer #2: **Yes: ** Dr. Anna Gergely

Reviewer #3: No

---

## [Author Response · Author response to Decision Letter 1]

28 Feb 2025

Dear Dr. de Lourdes Peña Garay,

Thank you for offering us the opportunity to revise the manuscript.

The reviews were exceedingly thoughtful and constructive. The reviewers raised excellent points, especially in regard to some of our analyses and interpretations which prompted us to learn and implement a new method. We’ve made major revisions to the paper addressing every point raised in the review, and as a result, the paper has been much improved. We reply individually to all the reviewers’ comments in the following pages.

Reviewer 1 Comments:

1. The authors also provide a detailed explanation of inter-coder reliability, noting the average across four rounds of coding. Given that the practice of averaging reliability has been debated, it would be helpful if the authors also reported the range to give a fuller picture. For example, the reliability for most categories is strong, though it is lower for "Conversational Basics" (0.55). Further explanation on why this category’s reliability is comparatively low and how this may affect the interpretation of results would be valuable.

Response: Thank you for these points and suggestions. We have added the interrater reliability range for each of the categories for which calculating the range was available We also note that vocal play and reading had perfect interrater reliability because none of the coders identified any instances of those categories within the subset of clips for which reliability was assessed so range is not reported for those two categories. As for conversational basics, this was a broad category that included greetings, backchanneling, polite phrases, exclamations, etc. This is one of the categories in which future research might benefit from dividing the category into subcategories. We have added this note about the conversational basics category to the manuscript and have elaborated on the relevant portion of the Limitations section of the Discussion.

2. There are a few areas that require further clarification. For instance, it is not entirely clear if "3,727 adult speech audio clips" refers to distinct utterances. Later, the authors mention “3,607 adult speaker utterances (2,210 IDS; 1,397 ADS) after exclusions,” suggesting that all clips were treated as individual utterances. How were utterances determined? In the OSF repository, the first clip (comfort_inform_CDS) includes two grammatical segments (clauses). Were they counted as one utterance?

Response: Thank you for bringing this to our attention. Adult speech audio clips were automatically identified by LENA software and varied in duration (by seconds). Therefore, a clip could include more than one utterance. Where applicable in the manuscript, we changed “utterances” to “clips” and added a sentence in the Dataset subsection to explain that clips could contain more than one utterance.

3. The pragmatic context coding scheme for utterances includes categories such as "Inform," "Conversational Basics," "Question," "Imperative," "Reading," "Singing," "Comfort," and "Vocal Play." It would be helpful to know whether all utterances fit into these categories or if there were cases that could not be coded within this framework.

Response: Thank you for the suggestion. We now report the number of clips that were not tagged with a pragmatic context toward the beginning of the Results section.

4. Beyond the factors analyzed, I am curious if the authors considered other potential influences. For example, was the time of day when utterances occurred recorded, and could this factor influence speaker presence by gender? Additionally, given that 43 infants had parents with higher education, could parental education also play a role in prosodic modification patterns?

Response: Thank you for these suggestions. We did not analyze other potential influences like time of day or parental education. The IDSLabel dataset and metadata do not include time of day, but it would be an interesting factor to consider and could potentially be pursued by linking the IDSLabel data back to the original HomeBank corpora from which they derive. We have added these suggestions to the Discussion section.

5. Further detail on the perceived gender distribution of adult speakers within each participant and across the total sample is recommended. Providing descriptive information about these characteristics would offer a clearer picture of the sample's demographic composition. It would also be beneficial if the authors elaborated on the automatic pitch detection methodology and whether they validated this method against human-coded data.

Response: Thank you for the recommendation. In the Dataset section, we have added descriptive information about the perceived gender of adult speakers across the total sample. The participant-level perceived adult gender information is available within the IDSLabel dataset and the materials that are shared on our GitHub repository, linked within the OSF project associated with the manuscript. Regarding the pitch analyses, Dr. Kyle MacDonald, the first author of the paper from whom the pitch values were obtained is no longer working in academia. The current paper’s senior author served as his mentor on that project and according to her memory and notes, the values were chosen based on informal pilot explorations informed by the soundgen manual and references to prior studies applying automated pitch estimation to caregiver speech samples. Formal validation against human-coded pitch contours was not performed. However, since the sound files are available on HomeBank, a motivated reader could attempt such a project. We did not update the methods to add these details because they are vague and based on personal communications and memories rather than details provided in the original paper and its associated GitHub repository (but we could add these comments to the manuscript if that seems appropriate and desirable). We did add a statement to the relevant section of the Discussion: “In the future, it would be helpful to validate the automatically obtained pitch estimates against pitch measurements obtained by expert humans.”

Reviewer #2 Comments:

1. I am not familiar of LENA device and its working mechanism. On the official website I read that it’s deleting the actual conversation and store only the acoustic data in order to meet personality rights. If so, how audio playbacks were coded by the coders? The website also stated that the device is not able to “understand” spoken words but it can count the words. But it can also count utterances (as in the article there are information about the number of utterances). In the article there are also utterance-level acoustic characteristics, so it seems that the device is able to do utterance-level analyses. Is it able to do hyperarticulation analyses? Or any other vowel-level analyses? Anyway, it would be expedient to write a little bit more about the device, the software, the output, etc. for those who are not familiar with this data acquisition method.

Response: Thank you for letting us know that this was confusing! The LENA system comes in different versions. The version that was used for all the recordings in the IDSLabel dataset was LENA Pro, which does save the audio and allows the user to export a WAV file for each recording. We have added some of this information to the Methods to hopefully make it clearer for readers. We also have changed to using the term “clip” instead of “utterance” as the latter can have linguistic definitions that don’t exactly map onto how the segmentation is done by the LENA software. We’ve added some more information about that segmentation process as well as a reference to a document written by LENA Foundation scientists that we hope helps clarify this. To answer your last two questions, no, the LENA software does not include hyperarticulation or other vowel-level analyses. The LENA resource just mentioned and added to the paper’s citations/references should clarify this for any interested readers who have the same question. We agree that those would be excellent to be able to include in future work, and we’ve now added vowel formants to the examples of other acoustic measures that could be included in future research.

2. It would be also nice to have a table with the number of clips in each pragmatic context category (authors noted that Reading and Vocal play were underrepresented and sometimes these contexts were excluded from analysis, but it would be important to see the exact numbers within each pragmatic context). It would be also important to see the exclusion rate of clips within each pragmatic context. For instance, we can read in the discussion that authors were surprised that comfort was rare in the dataset. But isn’t it possible that most of the comfort context had been excluded because the infant was crying and as it was considered as “noisy”? A detailed table about excluded clips would give an answer to this question.

Response: Thank you for bringing up this idea. The only clips that were excluded from our dataset is if they were only coded as noisy, or indecipherable (n = 107). And it is possible that some comfort may have been mistakenly categorized as “noisy”, as well as possible that some comfort may have not been included within IDSLabel, if it overlapped with child crying and was labeled as such by LENA. We have included this note in our discussion section. We have also included a table in our supplemental material detailing the number of clips in each category and the number of clips with a context code and a noisy code for further transparency.

3. Line 252 criterion of at least 20 utterances seems a little bit arbitrary. What was the basis for this inclusion criteria?

Response: Thank you for pointing out that this could be better explained. We’ve added the following statement: “This criterion was somewhat arbitrary and was based on our intuitions about what would be a reasonable minimum sample size for clip-level acoustic analyses; our main aim in specifying this criterion ahead of time was to avoid data dredging.”

Reviewer #3 Comments:

1. The authors made commendable efforts to address the issue of zero counts in some pragmatic contexts by introducing a randomization-based approach to include these categories in their logistic mixed-effects model. However, the limited observations for contexts such as 'reading,' 'singing,' and 'comforting,' particularly in the ADS register, raise concerns about the robustness of the conclusions drawn for these categories. Sparse or absent data for certain pragmatic contexts inherently limit the statistical power and reliability of the model's estimates for these categories. The artificial reclassification of data to enable model convergence, while creative, introduces additional uncertainty and potential biases that complicate the interpretability of the findings for these specific contexts.

Response: We are grateful for this criticism. In researching more about the possibility for statistical bias with small sample sizes, we discovered that taking a Bayesian rather than a maximum likelihood approach would be better for our logistic regression testing for differences in the prevalences of contexts in IDS vs. ADS. This change in approach enabled us to include all contexts in the model without the artificial random reclassification. (We did need to reduce the model complexity in order to achieve convergence, taking out adult gender as a predictor, but that variable was not significant with the prior approach.) This is one of those wonderful cases where a reviewer comment prompted us to learn and implement a new method, making both the current study and our own skillsets stronger.

2. The results from running the logistic mixed-effects model provided in the manuscript's accompanying code revealed no significant main effect of the pragmatic context of questions on the register, contrary to the findings reported in the paper (see attached html table). This discrepancy reinforces concerns about the robustness of the analysis, particularly given the limited observations for some pragmatic contexts and the reliance on resampling techniques to address zero counts. Furthermore, the warning message generated during model execution—“Model is nearly unidentifiable: large eigenvalue ratio - Rescale variables?”—raises additional doubts about the model's stability and reliability.

Response: Thank you so much for taking the time to run our models and for discovering this discrepancy. We must apologize; the previous submission included a typo in the reporting of the questions result from this analysis, accounting for the discrepancy. As we are now using a Bayesian model in place of the maximum likelihood based model used previously, small sample size is less of a concern (see van de Schoot et al., 2014), and the Bayesian regression runs without warnings and with good diagnostics (https://mc-stan.org/learn-stan/diagnostics-warnings.html).

van de Schoot, R., Kaplan, D., Denissen, J., Asendorpf, J.B., Neyer, F.J. and van Aken, M.A.G. (2014), A Gentle Introduction to Bayesian Analysis: Applications to Developmental Research. Child Dev, 85: 842-860. https://doi.org/10.1111/cdev.12169

3. The study analyzes IDS across a broad age range of 3 to 20 months, which spans a critical period of infant development. Previous research has demonstrated that IDS evolves as infants grow, with notable changes in pitch characteristics (e.g., Cox et al., 2023), and pragmatic functions tailored to developmental stages, such as capturing young infants’ attention and conveying emotional affect versus supporting linguistic goals in older infants (Fernald, 1992; Kuhl et al., 1997). By aggregating data across this wide age range without accounting for potential developmental differences, the study may obscure important age-specific patterns in IDS. For example, pitch characteristics or the frequency of certain pragmatic contexts (e.g., comforting vs. reading) may differ markedly between younger and older infants, which could confound the results. A more nuanced analysis that examines IDS characteristics by narrower age bands or includes age as a covariate could provide deeper insights into how pragmatic contexts and pitch modulations vary across development.

Response: Thank you for these thoughtful comments and connections to prior literature. We completely agree that age could moderate IDS acoustics in a way that might differ across contexts and/or speaker gender and/or infant gender. We had originally considered including age in our analyses but decided not to do so due to concerns about statistical power and about model complexity posing a challenge to interpretation (we wanted to avoid 3-way interactions). We hope that future research can incorporate age as well as other factors, but we expect that some other statistical approach (e.g., random forest) may become necessary. For these reasons we have not attempted to add this to the current study, but we have expanded the discussion to include some of these points.

4. Furthermore, the inclusion of Spanish utterances in the dataset introduces an additional variable that could influence the pitch characteristics of IDS, as cross-linguistic research has shown that IDS varies significantly across languages (e.g., Fernald et al., 1989). For instance, Spanish IDS may exhibit different pitch ranges, prosodic contours, or modulation patterns compared to English IDS due to phonological and cultural differences in speech patterns (e.g., Cox et al., 2023). These linguistic variations could confound the analysis, particularly if the distribution of Spanish and English utterances is uneven across pragmatic contexts, speaker genders, or infant age groups.

Response: We agree that IDS intonation appears to have both similarities and differences across languages (as is conveyed in the Discussion). However, the number of clips in which the adults spoke Spanish was an extremely small proportion of the IDSLabel dataset. We thus think it very unlikely that this is a confound for our study. We have made some edits to the Methods where it is mentioned that there were a few Spanish clips, including adding the exact number so that readers can see that this was an extrem

---

## [Decision Letter · Decision Letter 1]

PONE-D-24-37730R1Pitch characteristics of real-world infant-directed speech vary with pragmatic context, perceived adult gender, and infant genderPLOS ONE

Dear Dr. Neer,

Thank you for submitting your manuscript to PLOS ONE. After careful consideration, we feel that it has merit but does not fully meet PLOS ONE’s publication criteria as it currently stands. Therefore, we invite you to submit a revised version of the manuscript that addresses the points raised during the review process.

We look forward to receiving your revised manuscript.

Kind regards,

Marcela de Lourdes Peña Garay, Ph.D

Academic Editor

PLOS ONE

Journal Requirements:

Reviewers' comments:

Reviewer's Responses to Questions

**Comments to the Author**

1. If the authors have adequately addressed your comments raised in a previous round of review and you feel that this manuscript is now acceptable for publication, you may indicate that here to bypass the “Comments to the Author” section, enter your conflict of interest statement in the “Confidential to Editor” section, and submit your "Accept" recommendation.

Reviewer #1: All comments have been addressed

Reviewer #3: (No Response)

2. Is the manuscript technically sound, and do the data support the conclusions?

Reviewer #1: Yes

Reviewer #3: Partly

3. Has the statistical analysis been performed appropriately and rigorously? 

Reviewer #1: Yes

Reviewer #3: Yes

4. Have the authors made all data underlying the findings in their manuscript fully available?

Reviewer #1: Yes

Reviewer #3: Yes

5. Is the manuscript presented in an intelligible fashion and written in standard English?

Reviewer #1: Yes

Reviewer #3: Yes

6. Review Comments to the Author

Reviewer #1: I appreciate the opportunity to review this revised manuscript. The authors have adequately addressed all of my previous comments.

Reviewer #3: I appreciate the thoughtful and thorough revisions made in response to the initial feedback. The authors have substantially improved the manuscript by increasing methodological transparency and adopting a Bayesian modeling approach to address issues related to sparse data. The updated analyses are clearly reported, and the convergence diagnostics suggest a well-specified model.

The authors note that adult speaker gender was removed from the Bayesian model to achieve convergence—an understandable decision given the data limitations and model complexity. However, since adult gender was central to the study’s original hypotheses and included in the preregistration, it would be helpful to explicitly acknowledge this as a deviation from the preregistered analysis and briefly reflect on its implications for interpreting the results. In addition, I recommend that the authors clearly state that the adoption of the Bayesian approach also constitutes a deviation from the preregistration (Variables and Analyses 1).

Reproducing the Bayesian analysis requires substantial computational time and resources (taking 11 hours for the Bayesian model to run on my end!). This could pose a barrier to reproducibility for researchers without access to high-performance computing setups. I suggest the authors consider caching model outputs (e.g., using saveRDS() or storing fitted model objects) and including them in the repository. This would facilitate transparency and reproducibility without requiring each user to rerun time-intensive sampling.

Minor:

p. 16: max_treedepth is 15 within the code but reported as 12 in the manuscript

Scaling and centering of binary predictors such as adult speaker gender, child gender, and register are performed using the scale() function. I am wondering whether effect coding (e.g., −0.5 / 0.5) would typically offer a more interpretable and robust approach in this context?

7. PLOS authors have the option to publish the peer review history of their article (what does this mean? ). If published, this will include your full peer review and any attached files.

**Do you want your identity to be public for this peer review?** For information about this choice, including consent withdrawal, please see our Privacy Policy .

Reviewer #1: No

Reviewer #3: No

---

## [Author Response · Author response to Decision Letter 2]

16 May 2025

Dear Editors,

Thank you for offering us the opportunity to revise the manuscript. We are grateful to the Editor and all the reviewers for their helpful comments. Below are responses to the comments and suggestions on the previous revision.

Reviewer #3 Comments:

1. The authors note that adult speaker gender was removed from the Bayesian model to achieve convergence—an understandable decision given the data limitations and model complexity. However, since adult gender was central to the study’s original hypotheses and included in the preregistration, it would be helpful to explicitly acknowledge this as a deviation from the preregistered analysis and briefly reflect on its implications for interpreting the results. In addition, I recommend that the authors clearly state that the adoption of the Bayesian approach also constitutes a deviation from the preregistration (Variables and Analyses 1).

Response: Thank you for this suggestion, and we agree that transparency is needed. We have added a statement in the “Preregistered Planned Analyses and Pilot Analyses” section to highlight the adoption of the Bayesian approach and that this approach differs from the preregistration. Additionally, we added a short paragraph in the discussion section on the implications of excluding speaker gender from the analysis.

2. Reproducing the Bayesian analysis requires substantial computational time and resources (taking 11 hours for the Bayesian model to run on my end!). This could pose a barrier to reproducibility for researchers without access to high-performance computing setups. I suggest the authors consider caching model outputs (e.g., using saveRDS() or storing fitted model objects) and including them in the repository. This would facilitate transparency and reproducibility without requiring each user to rerun time-intensive sampling.

Response: Thank you so much for taking the time to run our analyses and for this helpful suggestion. We have taken the suggestion to save the model object using saveRDS() and the file is now included in the GitHub repository associated with the manuscript. We also updated the results output text file on GitHub and have updated the numbers in Table 2 and the reported ESSs to match this most recent run of the model.

3. p. 16: max_treedepth is 15 within the code but reported as 12 in the manuscript

Scaling and centering of binary predictors such as adult speaker gender, child gender, and register are performed using the scale() function. I am wondering whether effect coding (e.g., −0.5 / 0.5) would typically offer a more interpretable and robust approach in this context?

Response: Thank you for catching our error in reporting the value of max_treedepth! The correction has been made in the manuscript. We also appreciate the suggestion to consider effect coding. While we did not make this change to the effect coding for this manuscript, as it would have been a fairly major revision. However, we will consider taking an effect coding approach in future studies.

---

## [Editor Report · Decision Letter 2]

Pitch characteristics of real-world infant-directed speech vary with pragmatic context, perceived adult gender, and infant gender

PONE-D-24-37730R2

Dear Dr. Neer,

We’re pleased to inform you that your manuscript has been judged scientifically suitable for publication and will be formally accepted for publication once it meets all outstanding technical requirements.

Kind regards,

Marcela de Lourdes Peña Garay, Ph.D

Academic Editor

PLOS ONE
---

## [Editor Report · Acceptance letter]

PONE-D-24-37730R2

PLOS ONE

Dear Dr. Neer,

I'm pleased to inform you that your manuscript has been deemed suitable for publication in PLOS ONE. Congratulations! Your manuscript is now being handed over to our production team.

Kind regards,

on behalf of

Dr. Marcela de Lourdes Peña Garay

Academic Editor

PLOS ONE